# The Correlation Between Body Pain Indicators and the Facial Expression Scale in Sows During Farrowing and Pre-Weaning: The Effects of Parity, the Farrowing Moment, and Suckling Events

**DOI:** 10.3390/ani15152225

**Published:** 2025-07-28

**Authors:** Elena Navarro, Raúl David Guevara, Eva Mainau, Ricardo de Miguel, Xavier Manteca

**Affiliations:** 1Department of Animal and Food Science, School of Veterinary Science, Universitat Autònoma de Barcelona, 08193 Bellaterra, Barcelona, Spainraul.guevara@awec.es (R.D.G.); xavier.manteca@uab.cat (X.M.); 2AWEC Advisors SL, Ed. Eureka, Parc de Recerca UAB, 08193 Bellaterra, Barcelona, Spain; 3Department of Animal Pathology, University of Zaragoza, 50013 Zaragoza, Zaragoza, Spain

**Keywords:** behaviour, facial expression, pain indicator, farrowing, suckling, parity

## Abstract

Giving birth is a painful experience for sows, yet research on their specific pain behaviours remains limited. The present study examined how pain indicators and facial expressions change during farrowing, during the farrowing event, and 19 days later. Sows displayed clear signs of pain, particularly at the moment of piglet expulsion. Back arching and facial tension were the most frequent body pain indicators, with significant differences depending on the stage of labour. After farrowing, these signs were absent. This research highlights a crucial issue in animal welfare: understanding and acknowledging pain in farm animals. By identifying reliable pain indicators in sows, farmers and veterinarians could improve birth conditions and introduce better pain management practices, ensuring a less distressing experience for the animals.

## 1. Introduction

Animal pain is a current problem in farm animals, which is receiving increased attention from scientists who are focusing on how to assess, control, and minimise it [1,2]. Pain in pigs can be caused by multiple factors, including some farm routine procedures, such as castration and tail docking [3,4] or physiological events, such as farrowing [5]. In particular, births associated with difficult parturitions or dystocia may cause pain in the mother [6,7]. It has been demonstrated that the administration of non-steroidal anti-inflammatory drugs (NSAIDs) after farrowing enhances sow recovery [8,9] and increasing piglet immunity transfer [10,11]. Recently, similar benefits have also been obtained by administering paracetamol [12] or *Cannabis sativa* [13].

Objective measurements of pain severity can be difficult, and it remains a critical issue in veterinary and biomedical research [2]. Pain assessment in animals tends to use three approaches based on the measurement of general body functioning, such as food and water intake or weight gain, physiological indicators, and behavioural indicators [14]. Both general body functioning and physiological indicators are often not pain-specific, which means they are often sensitive to other states of an animal rather than to pain (such as stress, illness, or fear…) [15]. Therefore, pain assessment based on behaviour has received increasing attention and is the most commonly used parameter to assess pain in farm animals [16].

The assessment of spontaneous pain-related behaviour (spontaneously occurring behaviours arising or increasing in the context of painful conditions) is one of the most promising methods for assessing pain in pigs [17,18]. Pain-specific behaviours have been studied in pigs after castration, such as tail wagging [19], tail docking (tail wagging and jamming), teeth clipping (teeth champing), and ear notching (head shaking) [20]. Facial expressions in piglets have been studied, resulting in them being a good tool for evaluating when they are suffering from pain due to castration or tail docking [3,4].

Regarding pain at farrowing in sows, Ison et al. [18] described potential behavioural pain indicators, including pulling the back leg forward, trembling, back arching, pawing, and tail flicking. There are also studies that have described postural changes in the sow during farrowing [18,21,22,23]. In fact, pain can modify sows’ postural changes, which can lead to a higher risk for piglet crushing [22,24]. Navarro et al. [25] published a facial expression scale in sows, in which five different Facial Action Units (FAUs) were found in the sow’s face that determine if an animal is experiencing pain at a certain moment during farrowing.

Recently, several automatic monitoring systems using computer vision have been developed to observe farrowing- and lactation-related behaviours in sows and their piglets, including pain-related behaviours, such as posture activity [26] and facial expressions [27,28]. There are some factors that can modify the incidence of pain indicators during farrowing, such as the duration of the farrowing, the sows’ parity, and the different moments during farrowing. In fact, longer parturitions are considered to be more painful and problematic than shorter ones [29]. In modern swine production, farrowing lasting more than 5 h should be considered longer than average and a risk for increased perinatal mortality [30]. During farrowing, multiparous sows show a higher number of pain-specific behaviours than primiparous sows, but the latter perform more postural changes than multiparous sows [31,32]. Navarro et al. [25] described that sows showed more facial expressions related to pain during piglet expulsions than in the moments between piglet expulsion, or in the post-farrowing moment, where the pain facial expressions were almost absent. To our knowledge, other factors, such as the effect of suckling events on pain experienced by sows during farrowing, are not documented. The aims of this study are: (1) to assess if behavioural pain indicators (both the facial expression scale and body pain indicators) are affected by the farrowing, parity, and suckling events, and (2) to determine the relationship between the facial expression scale and body pain indicators during farrowing. We expected that all of these behavioural indicators studied could change similarly during farrowing, and consequently, could be a good tool to assess pain in peri-parturient sows.

## 2. Materials and Methods

### 2.1. Animals, Housing, and General Management

The experimental procedure was carried out on a commercial farm (Casa Ramona, Sora, Barcelona, Spain) from May to June 2018 and is part of a doctoral thesis [33]. A total of ten Danbred sows, five primiparous and five multiparous sows in their second and third parity, were randomly selected on the day of parturition. The randomisation was conducted by placing a camera on sows that showed signs of imminent farrowing. The sows that showed signs of lesions, illness, or lameness, a BCS lesser than two, and the sows that farrowed during the night were not included in the study. The BCS was visually determined according to a scale of one (very thin) to five (obese) points. For each sow, the following productive parameters were obtained: total piglets born per litter (including born alive, stillborns, and mummified foetuses) and cross-fostered piglets per litter.

On day 109 of gestation, the sows were moved to the farrowing room and were allocated to farrowing crates (1.95 × 0.60 m) built with steel bars, which were centrally positioned in farrowing pens (2.40 × 1.80 m). The farrowing pens were built with a metal-slatted floor for sows and a plastic-slatted floor for piglets. The farrowing pens had a metal heat pad at 36 °C and a heat lamp for the piglets during their first seven days of life. The temperature in the farrowing room was kept constant at approximately 22 °C, and the lights were on from 7:00 to 18:00 every day of the week. The sows were fed three times a day, and water was available ad libitum from drinkers.

On day 116 of gestation, farrowing was hormonally induced in multiparous sows, with 1 mL of Dalmazin^®^ (D Cloprostenol 0.075 mg/mL, FatroIberica; Barcelona, Spain). The farrowing date was estimated based on the date of the first insemination. Primiparous sows were not hormonally induced. Farrowing induction was an established management protocol on the commercial farm studied, implemented to improve farrowing planning and piglet care [34]. Litter size was standardised at 14–15 piglets by cross-fostering within 8 h post-farrowing. Cross-fostering was performed between the sows that farrowed on the same day, and extra piglets from large litters were transferred to nurse sows. The piglets were weaned at 21 days of age and moved to another barn on the farm, equipped with conditioned infrastructure for very young piglets.

### 2.2. Experimental Procedure

Throughout farrowing (from before the first piglet expulsion until the last piglet was born), the sows were recorded with an action camera (SK8 Urban, SK8 HD 4K; Porto, Portugal, which was installed on the crate at 120 cm above the floor where the sow was lying down. The camera was focused on the sow’s face, and its angle allowed for the recording of the rest of the body and the movements of the sow. This was necessary to guarantee complete visibility of the sow’s face and body. A vertical metal barrier (50 cm × 30 cm × 2 cm) was installed in the cranial part of the crate (in front of the sow’s feeder) to prevent the sow from putting her head under the feeder. If the sow was uncomfortable with the vertical metal barrier installed in front of the feeder (e.g., restlessness and difficulty lying down inside the crate), it was excluded from the study.

Nineteen days post-farrowing (two days pre-weaning) sows were recorded again for 2 h, between 8:30 h and 12:30 h, avoiding meals and farm management with the aim of obtaining videos as control moments. It was assumed that after more than two weeks of farrowing, there was no pain associated with this, or its intensity was minimal.

For each sow, the duration of farrowing, defined as the time between the first and the last piglet born and the time of each piglet expulsion, was registered by recorded-video direct observations.

### 2.3. Images, Selection, and Evaluation

One observer reviewed all the videos (partum and post-partum) and selected a total of 122 images when the sow was lying laterally, with her face visible and immobile. Blurry images or those in which the entire face was not visible were not selected. Once all the images were collected, they were edited so that only the faces of the sows were visible, to guarantee the blinding of the observer by not revealing the rest of the sow’s body and the presence or absence of piglets. The images were then randomly mixed and analysed independently and blindly by the observer, according to parity and the moment studied.

The three different moments selected were the following:Expulsion of the piglets (indicative of severe pain; n = 41 images). One image was chosen within the 30 s prior to each piglet expulsion.Inter-expulsion moment, described as the interval time between the delivery of two piglets (indicative of moderate pain; n = 43 images). One image was chosen at each interval.Pre-weaning (indicative of pain-free; n = 38 images). Images were chosen every 15–20 min.

The following five FAUs in the sows were studied, based on Navarro et al. [25]: tension above eyes, snout angle, neck tension, temporal tension and ear position, and cheek tension. Even if the neck is not a part of the face, the tension of this body area is influenced by the movements of the FAUs. The observer scored all the FAUs in each image and assigned a degree of pain: 0 (pain-free), 1 (moderate pain), 2 (severe pain), and IDK (“I don’t know”, meaning “I do not know or I do not feel confident assigning a pain score”). Additionally, a Total Facial Index (TFI) was calculated as the sum of the 5 FAUs, obtaining a value that ranged from 0 (pain-free) to 10 (maximum degree of pain) and attributing a global facial expression score to each image.

### 2.4. Video Analysis

A total of 10 sows and 56 h of video were studied. All videos (including farrowing and pre-weaning) from the same sows studied were observed continuously by the observer, who was masked by parity. Behavioural observations of the BPIs, sow postures, and suckling events were continuously performed in the three above-mentioned moments (the expulsion of the piglets, the inter-expulsion moment, and pre-weaning) (Table 1). For this study, behavioural pain indicators described by Ison et al. [17] are referenced as body pain indicators (BPIs), because both facial and body indicators are considered behavioural indicators.

### 2.5. Statistical Analysis

All statistical analyses were performed using IBM SPSS 19.0 for Windows (IBM Corp.; Armonk, NY, USA). Significance was set at *p* < 0.05, and a tendency was considered at *p* < 0.1 in all cases.

A Mann–Whitney Wilcoxon test was used to test whether the sows’ performance data were significantly different between parities (primiparous vs. multiparous). Graphs were produced with Prism 8.0.2 (GraphPad Software). The BPIs were analysed and represented as the number of events per minute. The FAUs were calculated as the mean value of the pain score (0, 1, 2) assessed in the images of the three moments studied. Additionally, total BPIs (as the sum of the 5 BPIs) and a TFI (as the sum of the 5 FAUs and scored from 0 to 10) were also analysed. For the BPIs and FAUs, general lineal models (GLMs) were used to assess the effect of different fixed factors, such as the farrowing moment (pre-weaning; inter-expulsion moment; and piglet expulsion), parity effect (primiparous vs. multiparous), and suckling event effect (piglets suckling vs. no piglets suckling). When the GLMs showed statistically significant differences, Duncan’s post hoc test was employed. The farrowing duration was correlated with each BPI and FAU using Spearman’s rank correlation coefficient (ρ). Similarly, the BPIs and FAUs were correlated amongst themselves. The interpretations of this ρ coefficient value were as follows: negligible (ρ < 0.20), weak (0.21 ≥ ρ < 0.40), moderate (0.41 ≥ ρ < 0.60), strong (0.61 ≥ ρ < 0.80), and very strong (0.81 ≥ ρ < 1.0). The sows’ posture (L.Lat, L.St, St, Sit) and postural changes (Yes vs. No) were analysed based on the parity and the farrowing moment. Assessment of the association between these discrete variables was carried out using Pearson’s Chi-square test or, alternatively, the Likelihood ratio and Fisher’s exact test, when needed. When this test showed statistically significant differences, standardised residuals as a post hoc test were used after a Bonferroni adjustment of the alpha value.

## 3. Results

The results on performance at the individual sow level are summarised in Table 2. Both parities were similar regarding the total duration of farrowing and the number of productive piglets.

### 3.1. Incidence of Body Pain Indicators

The incidence of the total BPIs was significantly different between the three different moments studied (*p* < 0.001). The sows exhibited a higher number of BPIs per minute during the piglet expulsion moment (mean = 4.62 BPIs/min), followed by the inter-expulsion moment (mean = 1.92 BPIs/min). Both farrowing moments showed a higher number of BPIs than the pre-weaning one, where the incidence of BPIs per minute was infrequent (mean = 0.12 BPIs/min). Additionally, there was a parity effect, as multiparous sows showed fewer BPIs than primiparous ones during the pre-weaning (primiparous: 0.18 BPIs/min vs. multiparous: 0.06 BPIs/min; *p* < 0.001) and inter-expulsion moments (primiparous: 2.88 BPIs/min vs. multiparous: 1.02 BPIs/min; *p* < 0.001). No differences based on parity effect were found during piglet expulsion (primiparous: 4.68 BPIs/min vs. multiparous: 4.56 BPIs/min; *p* = 0.035). Suckling events did not affect the incidence of the total BPIs (*p* = 0.285).

The individual BPIs are shown in Figure 1. Five BPIs showed significant differences between the three moments studied (*p* < 0.001). The incidences of leg, paw, and arch movements were higher during piglet expulsion, followed by the inter-expulsion moment. At the pre-weaning moment, they were much more infrequent. The tremble BPI followed the opposite pattern during farrowing, with the movements being higher at inter-expulsion moments. Interestingly, only two out of the ten sows showed this body indicator. The tail BPI was only present during piglet expulsion (Figure 1).

Parity had a statistically significant effect on four BPIs (*p* < 0.001). Primiparous sows showed a higher incidence of these BPIs per minute than multiparous ones: tremble (pre-weaning: 0.003 vs. 0.000 BPIs/min; inter-expulsion: 0.860 vs. 0.037 BPIs/min; and piglet expulsion: 0.542 vs. 0.051 BPIs/min), leg (pre-weaning: 0.024 vs. 0.004 BPIs/min; inter-expulsion: 1.411 vs. 0.520 BPIs/min; and piglet expulsion: 1.633 vs. 2.110 BPIs/min), tail tickling (pre-weaning: 0.000 vs. 0.000 BPIs/min; inter-expulsion: 0.006 vs. 0.006 BPIs/min; and piglet expulsion: 0.151 vs. 0.191 BPIs/min), and paw (pre-weaning: 0.077 vs. 0.041 BPIs/min; inter-expulsion: 0.235 vs. 0.144 BPIs/min; and piglet expulsion: 0.920 vs. 0.485 BPIs/min) movements. However, arch movement showed no differences (*p* = 0.121) between multiparous and primiparous sows (pre-weaning: 0.076 vs. 0.025 BPIs/min; inter-expulsion: 0.385 vs. 0.327 BPIs/min; piglet expulsion: 1.469 vs. 1.721 BPIs/min). The total BPI was statistically significant only in the pre-weaning moment, with a higher number of BPIs during the suckling events, when piglets were suckling versus no piglets suckling (pre-weaning: 0.18 vs. 0.06 BPIs/min, *p* < 0.001; inter-expulsion: 1.98 vs. 1.56 BPIs/min, *p* = 0.936; and piglet expulsion: 4.62 vs. 4.74 BPIs/min, *p* = 0.761).

During the inter-expulsion moment, the farrowing duration tended to show a strong correlation with tail movement (ρ = 0.65; *p* = 0.078).

### 3.2. Facial Action Units

When the five FAUs were analysed globally, the TFI showed statistically significant differences when the three moments were studied (*p* < 0.001). The highest pain scores were noted during piglet expulsion (TFI = 8.97), followed by the inter-expulsion moment (TFI = 5.26). The lowest pain scores were significantly reported at the pre-weaning moment (TFI = 0.94). Parity had no statistically significant effects on the TFI (*p* = 0.813).

The individual FAUs are shown in Figure 2. Each individual FAU showed differences between the three monitored moments (*p* < 0.001). During the piglet expulsion moment, the sows showed a higher pain score than during the inter-expulsion moment, and that moment, in turn, showed higher scores than the pre-weaning moment (Figure 2). The temporal tension and ear position FAUs showed a statistically significant parity effect (*p* = 0.048), as multiparous facial expressions received higher pain scores than primiparous ones, especially at the inter-expulsion moment. Instead, there were no statistically significant changes between primiparous and multiparous sows, analysing tension above eyes (*p* = 0.612), snout angle (*p* = 0.944), neck tension (*p* = 0.636), and cheek tension (*p* = 0.456).

The farrowing duration did not correlate with the FAUs.

### 3.3. Correlation Between Body Pain Indicators and Facial Expressions

Spearman’s correlations among the five BPIs and the five FAUs showed that arch was the BPI with the highest positive correlation with the five different FAUs (Table 1). Individually, strong positive correlations were found between arch and tension above eyes, arch and snout angle, and arch and cheek tension. The paw and leg BPIs showed a moderate correlation, considering the five FAUs means. The correlation obtained by tail was weak, and tremble was negligible (Table 3).

Spearman’s correlation means of the total BPI and each individual FAU showed differences between each FAU. Tension above eyes, snout angle, and cheek tension showed a strong correlation with the total BPI (Table 3). Instead, neck tension, temporal tension, and ear position showed a weak-to-moderate correlation with the total BPIs (Table 3). As the tremble BPI was only present in two out of the ten sows studied, the correlation between the TFI and the “total BPI without tremble” was also documented. Overall, a strong correlation between the TFI and the total BPIs was obtained, with a moderate improvement when excluding the tremble BPI.

### 3.4. Activity During Farrowing

Statistically significant results in postural changes were detected among the three monitored moments (*p* = 0.001). The sows performed greater postural changes in the inter-expulsion moment and the pre-weaning one than in the piglet expulsion moment (*p* < 0.001 and *p* = 0.001, respectively) (Figure 3). Sow postural changes during farrowing showed statistically significant differences according to parity, as multiparous sows performed more postural changes per minute than primiparous ones (*p* < 0.001) (Figure 3).

The percentage of time spent by sows in each posture was different between the three assessed moments in both primiparous and multiparous sows (*p* < 0.001) (Figure 4). The lying lateral (L.Lat) posture was the most common posture in the three different moments, being much more frequent during farrowing, as compared with the pre-weaning moment. Accordingly, the sows spent more time lying sternal (L.St) and standing (ST) during the pre-weaning moment. Considering the parity effect, primiparous sows spent more time in the L.Lat posture than multiparous sows, and multiparous sows spent more time in the L.St posture than primiparous ones.

## 4. Discussion

In the present study, sow parity did not differ in terms of performance variables or the farrowing duration. This may be explained by the small sample size, the inclusion of only second- and third-parity sows in the multiparous group (which have a similar farrowing duration to primiparous sows [38]), and the absence of any problematic farrowing lasting more than 5 h [30].

### 4.1. Incidence of Body Pain Indicators Considering Parturition, Effect of Suckling Events, and Farrowing Moment

Four of the five BPIs studied showed the same pattern that Ison et al. [18] described: arching back, tail flicking, and pulling the back leg forward were much more frequent during farrowing, especially during piglet expulsion, being almost nonexistent during the inter-expulsion moment and after the farrowing process. Tail flicking is strongly associated with imminent birth, and it can be associated with localised somatic pain from the pressure of the newborn piglet on the pelvic structures surrounding the vagina [18]. Alternatively, tail flicking can be related to a physiological act associated with piglet expulsion [18]. Otherwise, we see this indicator during all of the farrowing, and it is positively correlated with the farrowing duration, where it is clearly seen that sows suffer pain.

In the present study, trembling showed great individual variability, as only two out of the ten sows studied performed it. Ison et al. [18] described trembling as a very common BPI during all of farrowing and argued that it is associated with pain, inflammation, and fatigue as parturition progresses. We agree that trembling is indicative of pain and inflammation. Yun et al. [39] assessed pain and stress in sows in an experimental procedure (insertion of an intravenous catheter) and correlated higher levels of plasmatic cortisol with trembling behaviour. But, in our study, trembling did not appear as an indicator of fatigue as parturition progressed, because the two affected sows started trembling almost at the beginning of parturition. Overall, we conclude that not all sows that are suffering during farrowing show the trembling indicator, but if trembling appears, it is indicative of pain and inflammation. However, given the very low number of animals observed, these findings should be interpreted with caution.

Ison et al. [31] studied the parity differences during the farrowing moment, showing that primiparous sows exhibited fewer back arches and fewer back-leg forward behaviours than multiparous sows. Contrarily, in this study, we found that primiparous sows performed more BPIs per minute than multiparous sows, particularly in the inter-expulsion moment. These results agree with the fact that primiparous sows generally experience more painful parturitions than multiparous ones [6,7,32]. Lack of experience of primiparous sows and/or management factors can partially explain these differences. It is important to consider that parity and farrowing induction were confounded in the present study, as multiparous sows were hormonally induced one day before the expected farrowing date (day 116, considering the average gestation length in Danbred of 117.6 days [40]), whereas primiparous sows were not. According to a meta-analysis [41], prostaglandins had no effect on the farrowing duration. Induction of parturition in sows three or two days before the expected farrowing date does not affect the farrowing duration or the risk ratio of stillbirth. Conversely, when induction is performed one day before the expected farrowing date, the stillbirth rate is reduced by 28% due to higher farrowing supervision and piglet care. In the present study, the farrowing duration was not affected by parity/farrowing induction and the expected reduction of stillbirth was not demonstrated, probably because both parity groups were equally supervised during the day. Consequently, the increased BPIs observed in primiparous sows cannot be attributed to the prolonged duration of farrowing and/or stillbirths, whereas other factors, such as lack of experience, appear to be important contributors to pain during farrowing [6]. A study using a balanced number of sows, varying in parity and farrowing induction, is therefore needed to clearly determine their exact influences on BPIs.

The only BPI that showed no differences between parities was arching the back. Arching can be related to uterine contractions and trying to adopt a posture that facilitates piglet expulsion. It has been documented that multiparous sows can experience more pain at the end of farrowing due to a loss of uterine tone compared to primiparous ones [31], but this was not demonstrated in the present study.

The effect of suckling events was statistically significant only in the pre-weaning moment, showing more BPIs when piglets were suckling. The intact teeth of piglets can hurt the teats of sows, which causes avoidance behaviours in the sows [42]. Avoidance behaviour indicates that the sow attempted to avoid piglets’ contact with their udders because of discomfort and frustration [43]. That effect is not observed during farrowing, presumably for at least two reasons. Firstly, the discomfort produced by piglet suckling can be lower at the beginning of lactation than at the end. Secondly, the discomfort produced by piglet suckling can be masked by the pain suffered during the farrowing.

### 4.2. Facial Expressions and Their Correlation with the Body Pain Indicators

The FAUs herein studied were developed in a previous study by Navarro et al. [25], and five of them showed good or very good reliability among different observers. In the present study, all the FAUs showed marked differences between the farrowing moments studied, and these results paralleled those obtained in the analysis of the BPIs. Maximum pain scores were found during piglet-expulsion moments, whereas the inter-expulsion moment was associated with moderate FAU pain scores, irrespective of the parity studied.

Focusing on the five FAUs’ Spearman’s correlations means, tension above eyes, snout angle, and cheek tension were the FAUs with a stronger correlation for four of the BPIs (leg, paw, tail, and arch). In Navarro et al. [25], these FAUs were also the ones with higher reliability between the observer and the farrowing moment. This positive correlation confirms the usefulness of the FAU’s pain scale and highlights it as a reliable and easy-learning tool that could be potentially applied in commercial farms. The parity effect showed moderate differences between the BPIs’ and FAUs’ results. A lack of differences between primiparous and multiparous sows could stem from the lower number of sows (and consequently, images) analysed, which limits the statistical analyses. Further studies applying the facial expression scale in a broader number of sows may help to clarify this point.

When studying the five BPIs and correlating them with the five different FAUs, arch was the BPI with the highest correlation with the five different FAUs. A strong correlation was found with tension above the eyes, snout angle, and cheek tension. Supporting our observation, Grégoire et al. [44] described the posture “arched back” to be a body indicator in lame sows, describing intense pain in them. The paw and leg BPIs showed a moderate correlation, taking into account the mean of the five FAUs because it could be that, as happened in the Ison et al. [18] study, both BPIs were present during all parturition, showing a higher incidence at piglet expulsion, but they look to be not different enough to show strong correlations. The tremble correlation was negligible, which could be due to the high individual variability in the trembling BPI, which consequently shows a low correlation.

### 4.3. Sows’ Postures, Postural Changes, and Time Spent in Each Posture

Results related to postural changes and total time spent in each posture agree with the previous research in crated sows [18,21,45]. Overall, the number of postural changes and time spent lying laterally increased on the day of parturition, as compared with the days after it. Lying laterally during the day of farrowing is associated with physical exhaustion after birth [46], being higher in primiparous sows. In the present study, we were able to identify that postural changes were mainly performed at the inter-expulsion moment instead of the expulsion moment, being higher in multiparous sows. In addition, lying sternal and standing behaviour appeared more frequently during the pre-weaning days, as compared to the farrowing day. Several days after farrowing, lying sternal is associated with sows often being awake, attentive, and able to interact with their piglets [46]. Alternatively, sows lying sternal (without showing teats for piglet suckling) can be associated with an avoidance behaviour due to pain or discomfort when piglets are suckling [42]. This avoidance behaviour is supported in our study by the increased BPIs in the pre-weaning moment that the sows showed when the piglets were suckling.

## 5. Conclusions

Sows’ BPIs are positively correlated with FAUs during farrowing. A higher prevalence of behavioural indicators is present when the sows experience stronger pain during the farrowing moment (especially at the piglet expulsion moment) than when the pain is not present (pre-weaning). The incidence of leg, paw, and arch, as well as other FAUs, appears to be a candidate indicator for detecting problems during farrowing at the farm level. Future steps should include testing their practicability by farmers and fully validating the indicators through the administration of a non-steroidal anti-inflammatory drug (NSAID) at farrowing or by comparing eutocic and dystocic farrowing.

## Figures and Tables

**Figure 1 animals-15-02225-f001:**
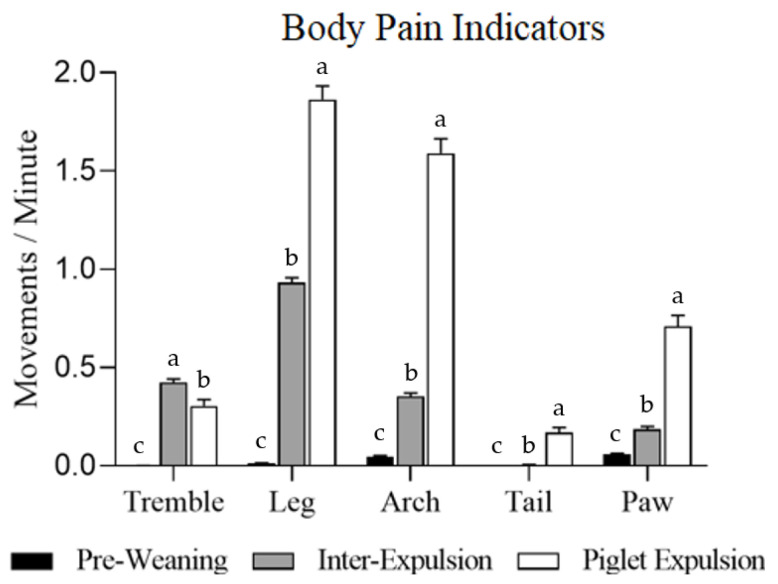
Individual body pain indicators per minute, expressed as the mean and standard deviation (SD). The bars represent the three moments studied: pre-weaning, inter-expulsion, and during piglet expulsion. Statistically significant differences (*p* < 0.001) between the three moments were found in the five body pain indicators, and represented with different letters (a, b, c).

**Figure 2 animals-15-02225-f002:**
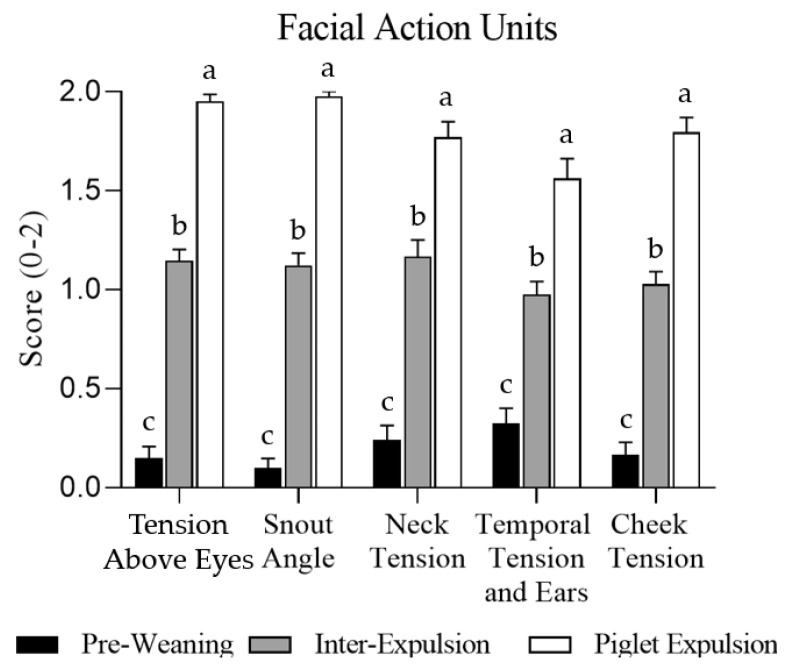
Sows’ Facial Action Units expressed in pain intensity punctuation (0–2) per image, represented as the mean and standard deviation (SD). The bars express the three moments studied: pre-weaning, inter-expulsion, and during piglet expulsion. Statistically significant differences (*p* < 0.001) were found in the five Facial Action Units when the farrowing moment was studied, and represented with different letters (a, b, c).

**Figure 3 animals-15-02225-f003:**
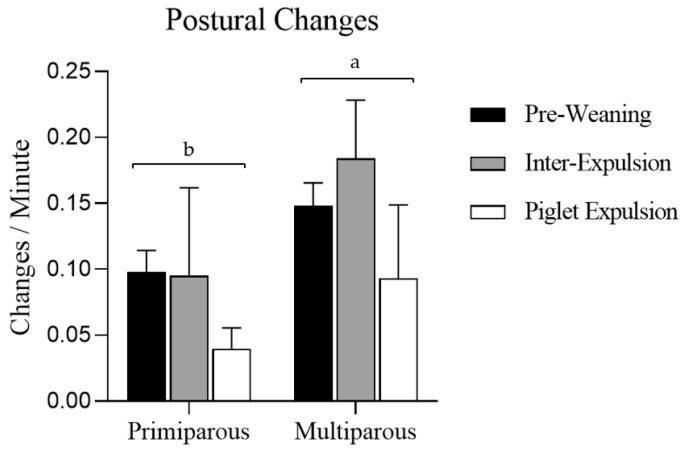
Sows’ postural changes per minute, expressed as the mean and standard deviation (SD), according to parity. The bars express the three moments studied: pre-weaning, inter-expulsion, and during piglet expulsion. Statistically significant differences (*p* < 0.001) were found when the parity was studied, and represented with different letters (a, b).

**Figure 4 animals-15-02225-f004:**
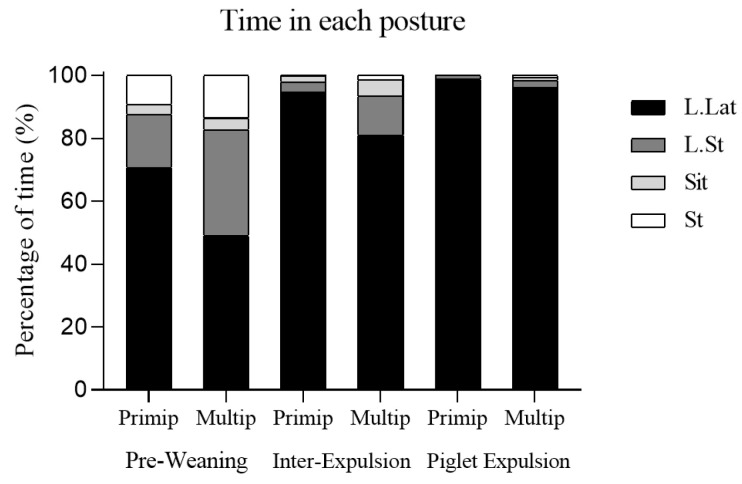
Percentage of the total farrowing time spent in each posture: lying lateral (L.Lat), lying sternal (L.St), sitting (Sit), and standing (St). The bars express the three moments studied: pre-weaning, inter-expulsion, and during piglet expulsion, and differentiate between multiparous and primiparous sows. Statistically significant differences (*p* < 0.001) were found when the moment was studied.

**Table 1 animals-15-02225-t001:** Ethogram of the behaviours continuously observed during the pre-weaning, inter-expulsion, and expulsion of the piglets.

Behaviours	Description
**Body Pain Indicators (BPIs)**	
Leg	The back leg pushes towards the abdomen [18]
Paw	Front leg is strongly pulled forward [18]
Arch	Back arching in lateral lying position, one or both sets of legs are pushed away from the body and/or inwards towards the centre [18]
Tremble	Visible shaking of the sow’s body [18]
Tail tickling	Tail is moving intensely in all directions [18]
**Postures**	
Lying lateral (L.Lat)	Lying on the side with the udder exposed [35]
Lying sternal (L.St)	Lying on the belly, with the front legs under the sow [35]
Standing (St)	An upright posture on extended legs, with all four feet contacting the ground [36]
Sitting (Sit)	The posterior end of the sow’s body in contact with/supported by the ground. The anterior of the body supported by the two front feet/legs [36]
Postural change	Transitions from lying to sitting or standing and reversed
**Suckling events**	
Piglets suckling	If one or more piglets are in contact with the udder or teat with their nose or mouth. Vigorous and rhythmic up and down head movements or suckling movements (Yes or No) [37]

**Table 2 animals-15-02225-t002:** The mean and standard error (SE) of the performance parameters and treatment records studied in the primiparous and multiparous sows during the whole trial period (from farrowing to weaning at 21 days).

	Primiparous (n = 5)		Multiparous (n = 5)		*p*-Value
Items	Mean	SE	Mean	SE	
Parity	1.0	0.00	2.8	0.2	
Total duration of farrowing (h)	3.1	0.34	3.2	0.44	0.764
Total piglets born per litter	15.8	0.58	18.6	2.32	0.275
Born alive per litter	14.6	0.24	16.8	2.18	0.345
Stillborns per litter	1.0	0.45	1.4	0.75	0.659
Mummified foetuses per litter	0.2	0.20	0.4	0.40	0.667
Cross-fostered piglets per litter	14.2	0.20	14.8	0.20	0.701

**Table 3 animals-15-02225-t003:** Spearman’s correlation between the five Facial Action Units studied and the five body pain indicators (BPI), including the total mean and the total mean without tremble.

Facial Action Units	Body Pain Indicators						
	Tremble	Leg	Arch	Tail	Paw	Total BPI	Total BPI Without Tremble
Tension above eyes	0.141	0.520 ***	0.737 ***	0.383 ***	0.591 ***	0.771 ***	0.799 ***
Snout angle	0.141	0.445 ***	0.709 ***	0.311 ***	0.590 ***	0.718 ***	0.746 ***
Neck tension	0.108	0.397 ***	0.578 ***	0.252 **	0.449 ***	0.568 ***	0.596 ***
Temporal tension and ear position	0.060	0.367 ***	0.543 ***	0.246 **	0.396 ***	0.515 ***	0.573 ***
Cheek tension	0.151	0.508 ***	0.622 ***	0.319 **	0.532 ***	0.711 ***	0.735 ***
Total Facial Index ^1^	0.116	0.529 ***	0.704 ***	0.332 **	0.558 ***	0.722 ***	0.763 ***

^1^ Total Facial Index = the sum of the 5 Facial Action Units (ranging from 0—painless to 10—maximum degree of pain). **** p* < 0.001, ** *p* < 0.01.

## Data Availability

Data available from the corresponding author on reasonable request.

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
