# Peer review of "The Correlation Between Body Pain Indicators and the Facial Expression Scale in Sows During Farrowing and Pre-Weaning: The Effects of Parity, the Farrowing Moment, and Suckling Events"

_animals, 2025, doi:10.3390/ani15152225_

Round 1
Reviewer 1 Report
Comments and Suggestions for Authors
In the objectives, you state, among other things, that you wanted to assess the impact of parity and suckling events on pain indicators, but this is not summarized in the abstract and conclusion.
In line 311, the citation Ison et al. (2016) needs to be edited.
Which of the monitored pain indicators appears to be the most suitable for a simple assessment of pain intensity in practical conditions?

Author Response
I appreciate the valuable feedback and suggestions provided. Following is the point-by-point response to the suggestions and comments made. Additionally, changes are shown in the file with the track changes. Authors' responses are shown as “AR” in red color.
In the objectives, you state, among other things, that you wanted to assess the impact of parity and suckling events on pain indicators, but this is not summarized in the abstract and conclusion.
AR: the information has been added as suggested (L31-33)
In line 311, the citation Ison et al. (2016) needs to be edited.
AR: The citation has been fixed as requested (L356).
Which of the monitored pain indicators appears to be the most suitable for a simple assessment of pain intensity in practical conditions?
AR: In the conclusion (see L463–465), we included the indicators that appeared to be candidate measures for detecting problems during farrowing at the farm level: incidence of leg, paw, and arch issues, as well as FAUs, because these indicators varied across the three different time points studied. Which of these will be the most practical requires further investigation.
Reviewer 2 Report
Comments and Suggestions for Authors
The article submitted for review is interesting and important from the point of view of expanding knowledge about animal (sows) welfare, however, in my opinion it contains the obvious statement that the pain associated with labour is greater at the moment of labour, than 19 days after birth.
Below are the doubts and questions that arose while reading the article:
1) One of the most important is whether the fact that some sows (multiparous) had an induced labor and the other (primiparous) had a natural labor did not affect the length of labor and the level of pain? The Authors selected only 10 sows for the study, which is a relatively small number, so it is surprising that they did not standardize these groups in terms of the lack of labor induction.
This is particularly important, because the Authors themselves indicated in the discussion (LL341-344) that labor induction reduces the risk of stillbirth by 28 percent, and thus can regulate the course of labor (including its duration!), and thus affect the feeling of pain. They also mention this in the Introduction section (LL79-80). In my opinion, this is a significant methodological error that should be strongly justified in the Material and Methods section.
2) There is lack of very important information about the litters size among the sows studied. Knowing that older sows usually have more numerous litters, their labor probably lasted longer, and according to the information included in the Introduction section (LL79-80) - longer labor = more signs of pain.
3) How was it assessed which sow felt comfortable and which felt uncomfortable (LL124-125)?
4) There are 4 hours between 8:30 and 12:30. Why were these hours chosen? Why was the recording made for 2 hours? It was assumed (rightly, in my opinion) that after more than two weeks the sow would no longer show signs of pain associated with labor. But isn't 2 weeks too long? And why was day 19 chosen and not, for example, 15 or 17? What were the reasons for choosing 19 days after labor (LL126-129)?
5) I don't quite understand why all the photos were mixed? Additionally, only 43 photos were selected for the "inter-expulsion moment". For 10 sows, that's an average of 4 photos per sow, and the litters were probably much larger. Unfortunately, with such a description, I don't quite understand the criteria for selecting photos and why only 122 (122 out of how many?) were selected.
6) I understand that the authors based their research on other studies, but why was the "neck tension" category assigned to FAU?
7) It is not clear to me how the paper assessed "suckling events" because the phrase "pre-weaning period/moment" appears throughout almost the entire paper. If I am not mistaken, the phrase "suckling events" appears only in the Discussion (L352). Are these phrases used interchangeably by the Authors? In my opinion, this is not entirely justified.
Additionally, I understand that the "pre-weaning period" was assessed only once, on day 19, for 2 hours. How much food was consumed by the piglets then? In this context, the title of the paper does not fully correspond to the content presented in the results.
Additionally, the number of piglets in a litter can definitely affect the competition between piglets, and thus udder injuries and the sow's feeling of pain, especially on day 19 after birth, when the piglets are already quite large and here the litter size can play a particularly important role (but there is no such information about the litter size in the paper).
Additional comments:
LL93-95 - in my opinion this should be in the MM section
L109 - "Three meals per day were offered to the sows..." better replace "Sows were feed three times a day..."
LL151-152 - the listed indicators should be written in lower case
LL220-221- "A total of 10 sows and 56 hours of videos were 220 studied." - this should be in the MM section
L228 - tail tickling
L253 - why does "orbital tightening" appear here and in Fig. 2 when previously "Tension above eyes" was mentioned everywhere as the fifth FAUs indicator? Tension above eyes is also listed in Tab. 1.
LL282-283 - "Statistically significant results in postural changes around farrowing in sows were found according to the moment studied (p = 0.001)." - in my opinion this is incorrect (and requires standardization and clarification in the entire work) because it is difficult to talk about the period "around farrowing" on the 19th day after birth, when the piglets are just before weaning from their mother.
LL294-295 - the same situation as before; in my opinion you can't write about "three farrowing moments" in a situation where one of them describes the 19th day after giving birth.
Author Response
We appreciate the valuable feedback and suggestions provided. Following is the point-by-point response to the suggestions and comments made. Additionally, changes are shown in the file with the track changes. Authors' responses are shown as “AR” in red color.
Comments and Suggestions for Authors
The article submitted for review is interesting and important from the point of view of expanding knowledge about animal (sows) welfare, however, in my opinion it contains the obvious statement that the pain associated with labour is greater at the moment of labour, than 19 days after birth.
AR: We used these three time points, which were assumed a priori to represent different pain intensities (pre-weaning = pain-free; inter-expulsion = moderate pain; expulsion of piglets = severe pain), to determine which of these pain-specific indicators (BPIs and FAUs) are most representative of pain. In this way, we can identify indicators that may serve as candidates for detecting problems during farrowing (indicators that should appeared around farrowing, but not during pre-weaning). The next validation steps include the use of NSAIDs or comparisons between eutocic and dystocic farrowing (see Conclusions, lines 463–465).
Below are the doubts and questions that arose while reading the article:
- One of the most important is whether the fact that some sows (multiparous) had an induced labor and the other (primiparous) had a natural labor did not affect the length of labor and the level of pain? The Authors selected only 10 sows for the study, which is a relatively small number, so it is surprising that they did not standardize these groups in terms of the lack of labor induction.
This is particularly important, because the Authors themselves indicated in the discussion (LL341-344) that labor induction reduces the risk of stillbirth by 28 percent, and thus can regulate the course of labor (including its duration!), and thus affect the feeling of pain. They also mention this in the Introduction section (LL79-80). In my opinion, this is a significant methodological error that should be strongly justified in the Material and Methods section.
AR: The present work was done in a commercial farm. This management (induction for multiparous vs. non-induction for primiparous) was not possible to change. The farmer had bad experience by syncronizing primiparous, so he didn’t want to do it again, and wants to synchronize multiparouns (on day 116) for farrowing planning, piglets managing (fostering), facility utilization and reproductive scheduling. We introduced it, as you recommended in the material and method section (L122-124).
You’re right, increased duration of farrowing, and farrowing inductions >3days before the expected farrowing date conduct to more painful and problematic farrowings. However, in the present study, farrowing duration and % of stillbirth did not differ between parities/induction groups. We’ve included additional discussion related to this issue (see L383-399).
- There is lack of very important information about the litters size among the sows studied. Knowing that older sows usually have more numerous litters, their labor probably lasted longer, and according to the information included in the Introduction section (LL79-80) - longer labor = more signs of pain.
AR: Agreed, production data and duration of farrowing have been included, although our sows didn’t show differences between parities. More specifically, productive data obtained (born alive, stillborn, mummified foetuses) and cross-fostered piglets were introduced in the material and methods (L108-110 and L196-197), results section (L218-220 and Table 1), and discussion (L345-349). Duration of farrowing was also included in the material and methods (L206-207), results (L218-220, Table 1, L268-269, L296) and discussion (L345-349, L360, L390-395).
- How was it assessed which sow felt comfortable and which felt uncomfortable (LL124-125)?
AR: When the front metal barrier was placed in front of the sows, if a sow changed posture very frequently or touched the rear barrier while lying down, the front metal barrier was removed and the sow was excluded from the study. This information has been added in L138–139.
- There are 4 hours between 8:30 and 12:30. Why were these hours chosen? Why was the recording made for 2 hours? It was assumed (rightly, in my opinion) that after more than two weeks the sow would no longer show signs of pain associated with labor. But isn't 2 weeks too long? And why was day 19 chosen and not, for example, 15 or 17? What were the reasons for choosing 19 days after labor (LL126-129)?
AR: From 8:30 to 12:30, there were no staff interventions (meals, feces cleaning, etc.). We video recorded for 2 continuous hours. Depending on the sow and camera availability, recording started at 8:30 or later, up to a maximum of 10:30, and finished between 10:30 and 12:30.
We were looking for a control period (a pain-free moment) to ensure that pain-specific indicators were low or absent. We chose 19 days after farrowing because it was the day furthest from parturition and before weaning when we could record without external human interventions. On day 20, the farm carried out several managements to prepare for weaning, and on day 21, the piglets were weaned.
- I don't quite understand why all the photos were mixed? Additionally, only 43 photos were selected for the "inter-expulsion moment". For 10 sows, that's an average of 4 photos per sow, and the litters were probably much larger. Unfortunately, with such a description, I don't quite understand the criteria for selecting photos and why only 122 (122 out of how many?) were selected.
AR: Agreed, we have rephrased our explanation of the image selection process (L148-154). The reason for the low number of images was that, for the assessment of the FAUs, we required sows to be in a lateral posture, completely immobile, and with the entire face visible. Video sections in which the animal was moving or the face was partly covered by the crate bars or the shadow of the drinker were not used for image selection. Therefore, we do not have discarded images, as these were simply not selected in the first place.
6) I understand that the authors based their research on other studies, but why was the "neck tension" category assigned to FAU?
AU: An explanation about the incorporation of the Neck tension as FAU has been added in the text (L175-176)
- It is not clear to me how the paper assessed "suckling events" because the phrase "pre-weaning period/moment" appears throughout almost the entire paper. If I am not mistaken, the phrase "suckling events" appears only in the Discussion (L352). Are these phrases used interchangeably by the Authors? In my opinion, this is not entirely justified.
AR: We assessed 3 moments: pre-weaning, inter-expulsion moment and expulsion of the piglets. In the 3 moment, suckling events were always registered/observed. We improved the presentation and grammar clarification to better understand this issue (see L32, L186, Table 1 for better definition of suckling events, L205, L236, L265, L350, L403)
Additionally, I understand that the "pre-weaning period" was assessed only once, on day 19, for 2 hours. How much food was consumed by the piglets then? In this context, the title of the paper does not fully correspond to the content presented in the results.
Additionally, the number of piglets in a litter can definitely affect the competition between piglets, and thus udder injuries and the sow's feeling of pain, especially on day 19 after birth, when the piglets are already quite large and here the litter size can play a particularly important role (but there is no such information about the litter size in the paper).
AR: We’ve included the number of piglets nursed per litter, which did not differ by parities (around 14 piglets/sow) (see Table 2). However, we don’t have information about food consumed by piglets on day 19. We have introduced the “pre-weaning” moment in the title.
Additional comments:
LL93-95 - in my opinion this should be in the MM section
AR: The sentence was moved to the M&M section to the L 188-190.
L109 - "Three meals per day were offered to the sows..." better replace "Sows were feed three times a day..."
AR: The sentence has been modified as suggested (L117)
LL151-152 - the listed indicators should be written in lower case
AR: The sentence has been fixed as suggested (L173-174)
LL220-221- "A total of 10 sows and 56 hours of videos were 220 studied." - this should be in the MM section
AR: The phrase has been moved to the M&M section (L183)
L228 - tail tickling
AR: The term has been corrected (L257)
L253 - why does "orbital tightening" appear here and in Fig. 2 when previously "Tension above eyes" was mentioned everywhere as the fifth FAUs indicator? Tension above eyes is also listed in Tab. 1.
AR: The correction in the sentence and in Figure 2 has been done (L286)
LL282-283 - "Statistically significant results in postural changes around farrowing in sows were found according to the moment studied (p = 0.001)." - in my opinion this is incorrect (and requires standardization and clarification in the entire work) because it is difficult to talk about the period "around farrowing" on the 19th day after birth, when the piglets are just before weaning from their mother.
LL294-295 - the same situation as before; in my opinion you can't write about "three farrowing moments" in a situation where one of them describes the 19th day after giving birth.
AR: Thanks for the suggestion on clarifying the term. The change has been made through all the manuscript (L29, 200, 227, 239, 248, 250, 273, 279, 293, 320, 328, 332 and 344)

Reviewer 3 Report
Comments and Suggestions for Authors
This article investigates pain in sows during farrowing, a topic that, surprisingly, hasn't been widely explored in science. The researchers aimed to understand if behavioral pain indicators (BPIs) are affected by the moment of farrowing, the sow's previous birthing experience (parity), and the presence of suckling piglets. Additionally, they looked for a relationship between Facial Action Units (FAUs) and these BPIs during farrowing.
This article is well-written and presents a clear and focused study on an important aspect of sow welfare. Research on the correlation between indicators of body pain and facial expressions, as well as the effects of farrowing, timing of farrowing, and lactation events, is highly relevant and contributes significantly. However, it would be interesting to include the indicator of farrowing duration, due to its importance in relation to parity.
Suggestions and comments:
Line 41- 94: The introduction is well-crafted and highly effective. It clearly establishes the background, justifies the research, reviews relevant literature, and precisely states the study's objectives. All these elements align perfectly with your title. Great job!
Line 99-101: Could the authors please clarify their methods for randomly selecting the sows and ensuring the animals were disease-free?
Line 111: Please clarify how the sows' estimated farrowing dates were determined?
Please note that while PGF2α is effective, if farrowing is induced too far in advance of the natural due date (e.g., more than 1-2 days), the sow's cervix and birth canal might not be fully prepared for the passage of piglets. This "unripeness" of the cervix can lead to more difficult and potentially painful farrowing.
Line 112: Could the authors explain why primiparous sows were not induced for farrowing?
Line 114: Please explain in detail what 'Litter size was standardized by cross-fostering within 8 h post-farrowing' (line 114) entails. Specifically, clarify if this involves only transferring piglets to a foster mother, or if the total litter size is also adjusted (e.g., by removing piglets or combining litters to reach a target number).
Line 130: I suggest including data related to the duration of farrowing in sows in your results. This variable, as indicated in your introduction, is a factor that can modify the incidence of pain indicators around farrowing.
Regarding the duration of farrowing (Lines 130, 196, 308, 404): I considered that farrowing duration is a significant factor influencing pain indicators. I suggest including:
Results Section: Please include the data and analysis related to farrowing duration.
Discussion Section: It's crucial to add a discussion on the farrowing duration variable, especially considering its potential impact on pain indicators as highlighted in the introduction.
Conclusions Section: Please incorporate the relationship between farrowing duration and your results, acknowledging potential differences based on parity."
Author Response
We appreciate the valuable feedback and suggestions provided. Following is the point-by-point response to the suggestions and comments made. Additionally, changes are shown in the file with the track changes. Authors' responses are shown as “AR” in red color.
Comments and Suggestions for Authors
This article investigates pain in sows during farrowing, a topic that, surprisingly, hasn't been widely explored in science. The researchers aimed to understand if behavioral pain indicators (BPIs) are affected by the moment of farrowing, the sow's previous birthing experience (parity), and the presence of suckling piglets. Additionally, they looked for a relationship between Facial Action Units (FAUs) and these BPIs during farrowing.
This article is well-written and presents a clear and focused study on an important aspect of sow welfare. Research on the correlation between indicators of body pain and facial expressions, as well as the effects of farrowing, timing of farrowing, and lactation events, is highly relevant and contributes significantly. However, it would be interesting to include the indicator of farrowing duration, due to its importance in relation to parity.
AR: Thank you for your advice. We have included performance data and farrowing duration. Additional statistical analysis has been done, and results discussed.
Suggestions and comments:
Line 41- 94: The introduction is well-crafted and highly effective. It clearly establishes the background, justifies the research, reviews relevant literature, and precisely states the study's objectives. All these elements align perfectly with your title. Great job!
AR: Thank you. We appreciate it.
Line 99-101: Could the authors please clarify their methods for randomly selecting the sows and ensuring the animals were disease-free?
AR: The randomization was based on placing the camera on sows that showed signs fo beginning farrowing, and excluding these sows with illness problems. We’ve rephrased this section (L103-107)
Line 111: Please clarify how the sows' estimated farrowing dates were determined?
AR: The estimated farrowing date was obtained from the farmer's records, based on the date of the first insemination (L120–121).
Please note that while PGF2α is effective, if farrowing is induced too far in advance of the natural due date (e.g., more than 1-2 days), the sow's cervix and birth canal might not be fully prepared for the passage of piglets. This "unripeness" of the cervix can lead to more difficult and potentially painful farrowing.
AR: Yes, agreed. Additional information about PGF2-alpha management has been included. Our multiparous sows were induced on day 116 (one day before the expected farrowing day). (please, see L384-385)
Line 112: Could the authors explain why primiparous sows were not induced for farrowing?
AR: The present work was done in a commercial farm. This management (induction for multiparous vs. non-induction for primiparous) was not possible to change. The farmer had bad experience by syncronizing primiparous, so he didn’t want to do it again, and wants to synchronize multiparouns (on day 116) for farrowing planning, piglets managing (fostering), facility utilization and reproductive scheduling (see L122-124). Additional discussion about this parity/induction confounded factor has been introduced (L380-396).
Line 114: Please explain in detail what 'Litter size was standardized by cross-fostering within 8 h post-farrowing' (line 114) entails. Specifically, clarify if this involves only transferring piglets to a foster mother, or if the total litter size is also adjusted (e.g., by removing piglets or combining litters to reach a target number).
AR: The protocol of cross-fostering has been introduced (L125-126)
Line 130: I suggest including data related to the duration of farrowing in sows in your results. This variable, as indicated in your introduction, is a factor that can modify the incidence of pain indicators around farrowing. Regarding the duration of farrowing (Lines 130, 196, 308, 404): I considered that farrowing duration is a significant factor influencing pain indicators. I suggest including:
Results Section: Please include the data and analysis related to farrowing duration.
AR: Duration of farrowing has been included in the material and methods (L206-207) and results (L218-220, Table 1, L268-269, L296). Similarly, productive data have also been introduced (material and methods: L108-110, L196-197; results: L218-220 and Table 1
Discussion Section: It's crucial to add a discussion on the farrowing duration variable, especially considering its potential impact on pain indicators as highlighted in the introduction.
AR: Discussion related to productive data and farrowing duration have been introduced (L345-349, L360, L390-395).
Conclusions Section: Please incorporate the relationship between farrowing duration and your results, acknowledging potential differences based on parity."
AR: As productive data and farrowing duration did not show differences between parity or induction groups, we have decided not to include this information in the conclusion section. Various reasons (e.g., low sample size, the multiparous group lacking sows with ≥5 parities, and/or induction of farrowing one day before the expected farrowing date) have been discussed previously but were not presented as general conclusions.

Reviewer 4 Report
Comments and Suggestions for Authors
Line 111-115: What was the criterion for deciding that only multiparous females were induced with prostaglandins?
All four figures in the document mention that significant differences were found. However, some type of symbols or letters should be included to indicate these differences.
While it is true that the conclusion suggests that further studies should consider the use of non-steroidal anti-inflammatory drugs, we know that on commercial intensive farms, this practice is not feasible due to management and economic considerations. In this regard, what would be a practical application of the results of the study?
Author Response
I appreciate the valuable feedback and suggestions provided by the reviewers. Following is the point-by-point response to the suggestions and comments made by the reviewers. Additionally, changes are shown in the file with the track changes. Authors' responses are shown as “AR” in red color.
Comments and Suggestions for Authors
Line 111-115: What was the criterion for deciding that only multiparous females were induced with prostaglandins?
AR: The present work was done in a commercial farm. This management (induction for multiparous vs. non-induction for primiparous) was not possible to change. The farmer had bad experience by syncronizing primiparous, so he didn’t want to do it again, and wants to synchronize multiparouns (on day 116) for farrowing planning, piglets managing (fostering), facility utilization and reproductive scheduling (see L122-124). Additional discussion about this parity/induction confounded factor has been introduced (L380-396).
All four figures in the document mention that significant differences were found. However, some type of symbols or letters should be included to indicate these differences.
AR: The figures were modified as suggested. Letters were added to point out the differences among the different assessed moments.
While it is true that the conclusion suggests that further studies should consider the use of non-steroidal anti-inflammatory drugs, we know that on commercial intensive farms, this practice is not feasible due to management and economic considerations. In this regard, what would be a practical application of the results of the study?
AR: We have included in the conclusion which indicators are most feasible and their practical application (see L460-462). We suggest using NSAIDs to fully validate the pain indicators, as several papers suggest their benefits in reducing pain (ref 8-11 in the introduction), although their economic viability has not been tested.

Round 2
Reviewer 2 Report
Comments and Suggestions for Authors
The authors have significantly improved the manuscript.
Two minor comments:
LL:150-157 - to maintain the chronology of the farrowing/rearing process, I suggest (but it is not mandatory) changing the order to: Expulsion of the piglets, Inter-expulsion moment, and finally Pre-weaning;
LL:325-329 - there is an editorial error regarding the font and the lack of a separate subchapter 4.1.
Author Response
The authors have significantly improved the manuscript.
Thank you very much for your help.
Two minor comments:
LL:150-157 - to maintain the chronology of the farrowing/rearing process, I suggest (but it is not mandatory) changing the order to: Expulsion of the piglets, Inter-expulsion moment, and finally Pre-weaning;
AR: Agreed, this is a more logical and clearer way to present the information. We have made this change throughout the manuscript.
LL:325-329 - there is an editorial error regarding the font and the lack of a separate subchapter 4.1.
AR: Agreed, arranged.
